# Dimerization of the β-Hairpin Membrane-Active Cationic Antimicrobial Peptide Capitellacin from Marine Polychaeta: An NMR Structural and Thermodynamic Study

**DOI:** 10.3390/biom14030332

**Published:** 2024-03-11

**Authors:** Pavel A. Mironov, Alexander S. Paramonov, Olesya V. Reznikova, Victoria N. Safronova, Pavel V. Panteleev, Ilia A. Bolosov, Tatiana V. Ovchinnikova, Zakhar O. Shenkarev

**Affiliations:** 1M.M. Shemyakin & Yu. A. Ovchinnikov Institute of Bioorganic Chemistry, Russian Academy of Sciences, 117997 Moscow, Russia; mironov@nmr.ru (P.A.M.); p.v.panteleev@gmail.com (P.V.P.); ovch@ibch.ru (T.V.O.); 2Interdisciplinary Scientific and Educational School of Moscow University “Molecular Technologies of the Living Systems and Synthetic Biology”, Faculty of Biology, Lomonosov Moscow State University, 119234 Moscow, Russia; 3Phystech School of Biological and Medical Physics, Moscow Institute of Physics and Technology (State University), 141701 Dolgoprudny, Russia; 4International Tomography Center, Siberian Branch of the Russian Academy of Sciences, 630090 Novosibirsk, Russia

**Keywords:** antimicrobial peptide, capitellacin, β-hairpin, dimerization, DPC micelles, thermodynamics, NMR spectroscopy

## Abstract

Capitellacin is the β-hairpin membrane-active cationic antimicrobial peptide from the marine polychaeta *Capitella teleta*. Capitellacin exhibits antibacterial activity, including against drug-resistant strains. To gain insight into the mechanism of capitellacin action, we investigated the structure of the peptide in the membrane-mimicking environment of dodecylphosphocholine (DPC) micelles using high-resolution NMR spectroscopy. In DPC solution, two structural forms of capitellacin were observed: a monomeric β-hairpin was in equilibrium with a dimer formed by the antiparallel association of the *N*-terminal β-strands and stabilized by intermonomer hydrogen bonds and Van der Waals interactions. The thermodynamics of the enthalpy-driven dimerization process was studied by varying the temperature and molar ratios of the peptide to detergent. Cooling the peptide/detergent system promoted capitellacin dimerization. Paramagnetic relaxation enhancement induced by lipid-soluble 12-doxylstearate showed that monomeric and dimeric capitellacin interacted with the surface of the micelle and did not penetrate into the micelle interior, which is consistent with the “carpet” mode of membrane activity. An analysis of the known structures of β-hairpin AMP dimers showed that their dimerization in a membrane-like environment occurs through the association of polar or weakly hydrophobic surfaces. A comparative analysis of the physicochemical properties of β-hairpin AMPs revealed that dimer stability and hemolytic activity are positively correlated with surface hydrophobicity. An additional positive correlation was observed between hemolytic activity and AMP charge. The data obtained allowed for the provision of a more accurate description of the mechanism of the oligomerization of β-structural peptides in biological membranes.

## 1. Introduction

The widespread use of antibiotics in medicine and livestock farming has led to the emergence of resistant bacterial strains, complicating the treatment of diseases [1]. In recent decades, the appearance of multidrug-resistant pathogens has become more frequent, creating a potential threat of new epidemics [2]. Natural membrane-active antimicrobial peptides (AMPs) promise an alternative way to fight pathogenic microorganisms. AMPs often have a high proportion of hydrophobic and cationic residues and, due to their amphipathicity and overall positive charge, can selectively interact with anionic bacterial membranes. The aggregation of the peptides into oligomeric pore complexes or ‘carpet’-like structures on the membrane surface disrupts the integrity of the membrane and causes bacterial cell death [3]. In accordance with the proposed mechanisms of membrane activity, the dimerization (oligomerization) of AMPs increases activity and selectivity toward bacterial membranes [4].

Among structurally diverse AMPs, β-hairpin peptides are attractive candidates for the development of new antibiotics [5]. They exhibit a broad spectrum of biological activity against bacteria [6], fungi [7], and viruses [8], along with anticancer [9] and antibiofilm activities [10]. The β-hairpin conformation of these peptides is usually stabilized by inter β-strand disulfide bond(s). Non-covalent dimerization in a membrane-mimicking environment has previously been observed for β-hairpin and related β-structural AMPs (e.g., protegrins from the pig *Sus scrofa* [11,12,13,14], arenicins from the polychaeta *Arenicola marina* [15], cathelicidin-1 (ChDode) from the goat *Capra hircus* [16], and thanatin from the insect *Podisus maculiventris* [17]) using NMR spectroscopy (Figure 1). These dimers are stabilized by intermolecular hydrogen bonds between the β-strands of different monomers, associating with each other in parallel or antiparallel arrangement, forming a joined β-sheet. It has been proposed that these β-structural dimers could associate further with the aid of lipid molecules, forming either toroidal pores or ‘carpet’-like structures in the bacterial membrane [11,15]. However, there are β-hairpin AMPs (e.g., the horseshoe crab tachyplesin [18] (Figure 1C), the PV5 mutant analog of horseshoe crab polyphemusin [19], protegrin-2 [20], and the [Val8/Arg] mutant analog of arenicin-1 [21]) that dimerize weakly or do not at all. This indicates the presence of a wide range of dimerization energies for different peptides. Moreover, an example of thanatin showed that the ability of peptides to dimerize can depend on the properties of the membrane-mimicking environment, for example, on the lipid charge [17,22].

Capitellacin is a small cationic (20 amino acid residues, two disulfide bonds, a net charge at the neutral pH of +5) β-hairpin AMP from the marine polychaeta *Capitella teleta* (Figure 1A). This peptide demonstrates antimicrobial activity against Gram-negative and Gram-positive bacteria, including drug-resistant strains [23]. A study on lipid bilayers showed that capitellacin induces membrane permeability like the structurally similar tachyplesin, but with slower kinetics [6]. Previously, we have shown that capitellacin adopts a twisted β-hairpin structure in an aqueous solution [23]. Two conformations of the peptide were observed due to the *cis–trans* isomerization of the Ser1–Pro2 peptide bond. In the present work, we found that capitellacin, in the membrane-mimicking environment of dodecylphosphocholine (DPC) micelles, was in exchange between monomeric and dimeric micelle-bound forms. Using high-resolution NMR spectroscopy, the spatial structures of two forms of capitellacin were determined, and the concentration and temperature dependencies of the peptide dimerization were studied. A comparison of capitellacin with other β-hairpin AMPs that form dimers in a membrane-like environments showed that dimer stability and hemolytic activity mainly depend on the hydrophobicity of the peptide, and the formation of dimers is determined by the association of the polar or weakly hydrophobic surfaces of the molecules. The identified structural and thermodynamic aspects of β-hairpin dimerization may be relevant to the processes occurring in biological membranes, which allows us to better understand the membrane-disrupting activity of capitellacin and similar peptides. 

## 2. Materials and Methods

### 2.1. Expression and Purification of ^15^N-Labeled Capitellacin

Capitellacin and its ^15^N-labeled variant were produced in a bacterial expression system as described previously [23]. The fusion protein containing His-tag, *Escherichia coli* thioredoxin A [Met37Leu], additional methionine residue, and the target peptide was expressed in *E. coli*, purified using Ni^2+^ affinity chromatography, cleaved by cyanogen bromide, and the target peptide was purified via reversed-phase HPLC. ^15^N-labeled capitellacin was expressed in M9 minimal medium supplemented with ^15^NH_4_Cl as a nitrogen source. The final yield was approximately 1.9 mg of ^15^N-labeled capitellacin per 1 L of the culture, a value that is 3 folds lower than that for the unlabeled peptide expressed in rich LB medium [23]. The molecular mass of ^15^N-labeled capitellacin (2416.04 Da) measured via MALDI-MS (Reflex III, Bruker Daltonics, Billerica, MA, USA) matched well with the theoretical [M+H]+ value of capitellacin with 37 ^14^N atoms substituted by ^15^N and two disulfide bonds formed (2416.16 Da, Appendix A). A 2D ^1^H-^15^N HSQC NMR spectrum of ^15^N-labeled capitellacin in water (Appendix A) showed good agreement of ^1^HN chemical shifts with those previously reported for the unlabeled peptide [23]. Two sets of backbone NMR signals were observed due to the *cis–trans* isomerization of the Ser1–Pro2 peptide bond. 

### 2.2. NMR Experiments and Data Analysis

The NMR study was performed using samples containing 0.8 mM of unlabeled or 0.15 mM of ^15^N-labeled peptide in aqueous solution (5% D2O, pH 5.4). Detergent (d38-DPC, CIL, Andover, MA, USA) was added to the samples using aliquots of a concentrated water solution until the desired detergent-to-peptide molar ratio (D:P) was reached. The NMR spectra were recorded on a Bruker Avance III 600 and 800 spectrometers equipped with cryoprobes (Bruker, Karlsruhe, Germany). ^1^H, ^13^C, and ^15^N resonance assignments were obtained following a standard procedure based on a combination of 2D ^1^H,^15^N-HSQC, ^1^H,^13^C-HSQC, TOCSY, NOESY, and 3D NOESY-^15^N-HSQC spectra using the CARA software (version 1.84, Zurich, Switzerland). 

The spatial structures of the capitellacin monomer and dimer in micelles were calculated in the CYANA 3.98 program [24] using NMR data collected at the 45 °C and D:P values of 200:1 and 100:1, respectively. Upper interproton distance constraints were derived from the intensities of cross-peaks observed in 2D and 3D NOESY spectra (τ_m_ = 100 ms) through a “1/r^6^” calibration. ^3^J_H_^N^_H_^α^ coupling constants were determined using amplitude-modulated ^15^N-TROSY spectra [15]. The ^3^J_H_^α^_H_^β^ coupling constants were estimated from the multiplet patterns in 2D TOCSY spectra. The secondary structure of capitellacin was calculated from ^1^H and ^13^C chemical shifts using TALOS-N [25]. The φ, ψ, and χ_1_ dihedral angle restraints and stereospecific assignments were obtained from the J-couplings, NOE, and TALOS data. Hydrogen bonds were introduced using the temperature gradients of amide protons (∆δ^1^H^N^/∆T) measured in the 20–45 °C temperature range. The standard distance restraints (implemented in CYANA) were applied to restrain the disulfide connectivity. Special restraints (implemented in CYANA) were used to support the symmetry of the dimer structure (an identical structure of the monomers).

To probe the topology of the capitellacin/micelle complexes, 12-doxylstearic acid (12-DSA, Sigma, Saint Louis, MO, USA) was added to the ^15^N capitellacin sample to a 12-DSA:DPC molar ratio of 1:50. The paramagnetic broadening of HN resonances was qualitatively estimated using signal intensities in the 2D ^15^N-HSQC spectra.

### 2.3. Accessing Codes

Experimental restraints, chemical shifts, and calculated 3D structures of capitellacin in DPC micelles were deposited in BMRB and PDB databases (IDs 34756 and 8B4R—monomer; IDs 34757 and 8B4S—dimer).

### 2.4. Analisys of Physicochemical Properties of β-Hairpin AMPs

The overall geometry of the capitellacin β-hairpin was analyzed as described in [26]. The molecular hydrophobicity potential (*MHP*) and hydrophobic surface area was calculated using the PLATINUM software version 1.0 [27]. The grand average of hydropathy (*GRAVY*) values according to the Kyte and Doolittle hydrophobicity scale [28] and other parameters were calculated using the ProtParam module of the Biopython package (https://github.com/biopython (accessed on 12 December 2023)). Data are presented as the mean ± SD. The number of samples (*n*) are indicated in the text.

The structures of the following dimeric β-hairpin AMPs were analyzed (see Appendix A for PDB codes): capitellacin in DPC micelles (present work), protegrin-1 in POPC bilayers [14], protegrin-3 in DPC micelles [13], arenicin-2 in DPC micelles [15], ChDode (Cetartiodactyla cathelicidin-1) in DPC micelles [16], and thanatin in LPS micelles [17]. The structures of the following monomeric β-hairpin AMPs were analyzed (see Appendix A for PDB codes): capitellacin in water [23] and DPC micelles (present work), protegrin-1 in water [29], protegrin-2 in DPC micelles [20], the monomer of protegrin-3 created by splitting the structure of the protegrin-3 dimer in DPC micelles [13], arenicin-2 in water [30], the [Val8/Arg] mutant of arenicin-1 in water [21], arenicin-3 in water [31], the covalent dimer of ChDode in water [16], cathelicidin-1 from the sperm whale *Physeter catodon* in water (PcDode) [16], thanatin in water [17] and DPC micelles [22], tachyplesin I in water and DPC micelles [18] , polyphemusin I in water [32], mutant polyphemusin I analog PV5 in water and DPC micelles [19], alvinellacin from the hydrothermal worm *Alvinella pompejana* in water [33], and gomesin from the spider *Acanthoscurria gomesiana* in water [34]. The structures of the following monomeric α-helical AMPs were analyzed (see Appendix A for PDB codes): chicken cathelicidin fowlicidin-1 in water/TFE (1:1) mixture [35], melittin from bee venom crystalized from aqueous solution [36], magainin-2 from the skin of the frog *Xenopus laevis* in DPC micelles [37], and synthetic magainin/melittin hybrid MSI-594 in LPS micelles [38].

To compare the non-specific membrane activities of different peptides, the minimal hemolytic concentrations (*MHCs*, the peptide concentration that induce 10% hemolysis in 2 h) were collected from different literature sources. *MHC* values for melittin (0.2 μM), arenicin-2 (4 μM), arenicin-1[Val8/Arg] (125 μM), ChDode (40 μM), PcDode (>128 μM), capitellacin (>128 μM), and tachyplesin I (25 μM) were taken from our comparative work (P.V.P., unpublished observations). *MHC* values exceeding a certain threshold (>128 μM) were arbitrarily set to 150 μM. *MHC* values for protegrin-1 (4 μM), thanatin (500 μM), polyphemusin I (25 μM), arenicin-3 (130 μM), and gomesin (150 μM) were recalculated from the values reported in [39], using tachyplesin as a reference. The *MHC* value for PV5 (50 μM) was assumed to be twice the *MHC* value for polyphemusin, as described in [40]. The *MHC* for fowlicidin-1 (2 μM) was taken from [41]. The *MHC* value for magainin-2 (300 μM) was recalculated from the value reported in [42], using tachyplesin as a reference. No direct data describing the hemolytic activity of MSI-594 were found, so we arbitrarily assigned this peptide the *MHC* value equal to that of magainin (300 μM). Log_10_(*MHC*) values were used to fit different combinations of the physicochemical parameters of the AMPs. The optimization of the fits was performed using the Solver MS Excel 2019 add-in.

### 2.5. Study of Capitellacin Dimerization

The dependence of the free energy of dimerization (∆G^D^) on the detergent concentration was measured at 30 °C and with a P:D in the range of 1:50–1:400. The temperature dependence of ∆G^D^ was measured at a P:D of 1:400 in the temperature range 5–50 °C. At each concentration and temperature point, a 1D ^1^H spectrum of unlabeled capitellacin was recorded using the ‘excitation sculpting’ water suppression scheme. The relative monomer and dimer concentrations were calculated by fitting the ^1^H^ε^ Trp19 line shapes to the sum of Lorentzian lines in Mathematica (Wolfram Research, Champaign, IL, USA). The signal integrals were corrected to account for transverse relaxation during the water suppression scheme. 

To analyze the ∆*G^D^* concentration dependence, the modified ‘micellar solvent’ model was used [43]. In this case, the equilibrium dimer dissociation constant (*K_d_*) can be expressed in the following form:(1)$$K_{d}=\frac{{\left[M\right]}^{2}}{\left[D\right]\cdot {\left[DetE\right]}^{\gamma }},\;\left[DetE\right]=\left[Det0\right]-Nm\cdot \left[M\right]-Nd\cdot \left[D\right]-CMC,\;\gamma =\frac{2\cdot Nm-Nd}{Ne}$$
where [*M*], [*D*], [*Det*0], and [*DetE*] are the concentrations of the peptide monomer, dimer, total detergent, and detergent forming ‘empty’ (peptide-free) micelles in the sample, respectively; γ is the apparent reaction order with respect to detergent or, in other words, the number of detergent molecules released during the dimerization reaction, expressed as a fraction of the empty micelle [44,45]; *Ne*, *Nm*, and *Nd* are the average numbers of detergent molecules (aggregation numbers) in the empty micelles and micelles containing the monomer or dimer of the peptide, respectively; and *CMC* is the critical micelle concentration of detergent (DPC). It was assumed that *Ne* = 55 [46] and *CMC* = 1.5 mM. 

This form of the equilibrium dissociation constant was used for consistency with the ∆*G^D^* values previously reported for other systems under standard conditions ([*Det*0] = 1M) [47] Indeed, if the [*Det*0] >> [*M*], [*D*], and *CMC*, then [*DetE*] ≈ [*Det*0], and Equation (1) is simplified to the following:(2)$$K_{d}=\frac{{\left[M\right]}^{2}}{\left[D\right]\cdot {\left[Det0\right]}^{\gamma }},\;\mathrm{or}\;\mathrm{at}\;[Det0]=1M,\;K_{d}^{std}=\frac{{\left[M\right]}^{2}}{\left[D\right]}$$

The free energy of the dimerization was obtained from the equilibrium dissociation constant with the following formula:(3)$$\Delta G^{D}=R\cdot T\cdot ln\left(K_{d}\right)$$

The enthalpy (Δ*H^D^*) and entropy (Δ*S^D^*) of the dimerization were calculated by fitting the temperature dependence of free energy with a linear equation:(4)$$\Delta G^{D}\left(T\right)=\Delta H^{D}-T\cdot \Delta S^{D}$$

## 3. Results

### 3.1. Dimerization of Capitellacin in DPC Micelles

Unlike the cytoplasmic membranes of eukaryotic cells, which predominantly contain zwitterionic lipids in the outer leaflet, the bacterial cell membranes contain a large proportion of anionic lipids, such as phosphatidylglycerol and cardiolipin. In addition, the outer membranes of Gram-negative bacteria and cell wall peptidoglycans of Gram-positive bacteria have a net negative charge due to lipopolysaccharide (LPS) or teichoic acid (TA) moieties, respectively [48]. It is believed that the specific binding of cationic AMPs to anionic bacterial membranes and cell walls is one of the important factors determining their antibacterial activity [3]. 

Solution-state NMR spectroscopy is the method of choice for the investigation of the 3D structures of membrane-active antimicrobial peptides [49]. However, this method does not permit the study of peptide structure directly in the lipid bilayer, and a suitable membrane mimetic is required. Small and fast tumbling detergent micelles are commonly used for NMR studies. However, due to their dynamic nature and high mobility, detergent micelles reproduce the charge distribution observed in real lipid bilayers very poorly. Therefore, zwitterionic detergents and especially dodecylphosphocholine (DPC) are traditionally used to study AMPs in NMR studies (see the references in the text). A DPC molecule has the same negatively charged phosphate group as phospholipids. Moreover, using an anionic detergent to study a cationic AMP can result in peptide aggregation, as shown in our recent study on ChDode in LMPG micelles [16]. To ensure consistency with previous studies, we also used DPC for the current capitellacin study.

Previously, we observed that capitellacin binds with high affinity to zwitterionic DPC micelles [23]. At the detergent-to-peptide molar ratio of 130:1, two micelle-bound forms of capitellacin were observed in the NMR spectra. These forms differed mainly in the chemical shifts of the residues of *N*-terminal β-strand [23]. Here, we confirmed these findings and observed that one of the forms (‘D’) was predominant at the low D:P values (e.g., 50:1, Figure 2A), while the population of the second form (‘M’) grew together with the D:P ratio (see the ^15^N-HSQC spectrum at a D:P of 200:1, Figure 2B). Based on these data and spectra qualities, two capitellacin samples with D:P ratios of 100:1 and 200:1 were used for structural studies, where the relative populations of the peptide forms were ~3:2 and 2:3 (D:M), respectively. To exclusively observe the signals of the D form, samples of ^15^N-labeled and -unlabeled capitellacin at a D:P of 50:1 were used (Figure 2A). 

At high D:P ratios, some signals of the D form exhibited additional splittings, with up to four smaller cross-peaks observed for some residues (Figure 2B, red circles). At low D:P ratios, the splittings were less pronounced, with only two cross-peaks observed (Figure 2A, red circles). These splittings are probably associated with the process of the *cis–trans* isomerization of the Ser1–Pro2 peptide bond. Probably, the rate and equilibrium of this exchange process in the capitellacin dimer depend on the D:P ratio. Indeed, the Ser1–Pro2 dipeptide belongs to the *N*-terminal β-strand responsible for the peptide dimerization (see below), and changes in this site can cause rearrangement in the dimer structure. For the symmetric capitellacin dimer, three structurally different states are possible (*cis/cis*, *cis/trans* (equal to *trans/cis*), and *trans/trans*). The presence of the fourth component of the signal can be explained if we assume that the dimer symmetry is broken (*cis/trans* is not equal to *trans/cis*).

The signals of form M also show some heterogeneity (Figure 2B, blue circle), but its multiplicity is lower than that of form D. Indeed, only two conformations (*cis* and *trans*) are possible for the peptide monomer. Interestingly, the isomerization of the Ser1–Pro2 peptide bond has little effect on the H^ε1^ signal of the Trp19 side chain (Figure 2A,B, cyan circle). This allows the integral intensity of the H^ε1^ signal to be used to monitor the monomer–dimer equilibrium (see below).

The almost complete ^1^H and ^15^N and partial ^13^C resonance assignments were obtained for both structural forms of capitellacin. The observed NOE contacts revealed the β-hairpin conformation in both forms. At the same time, for form D, we observed a subset of NOESY cross-peaks (Pro2:HA–Cys9:HN, Arg3:HN–Cys9:HN, Arg3:HN–Arg10:HA, Val4:HA–Arg7HN and Cys5:HN–Arg7:HN) that were incompatible with the monomeric β-hairpin structure. These contacts were consistent with intermonomer interactions in the symmetric dimer of capitellacin formed through the antiparallel association of *N*-terminal β-strands (Figure 1B and Figure 2D, green arrows and dot). An analysis of the NOESY spectra did not reveal ‘intermonomer’ contacts in the M form of the peptide. The temperature gradients of the amide proton chemical shifts (∆δ^1^H^N^/∆T) were also in agreement with the *N*-terminal dimerization of capitellacin in the D form. According to the ∆δ^1^H^N^/∆T data, the dimer was stabilized by additional (not observed in the monomer) H-bonds, formed by the HN of Cys5, Arg7, Cys9, and probably Arg3 (Figure 2C, stars). Thus, the NMR data confirm the assignment of the D and M structural forms of capitellacin in DPC micelles to a symmetric dimer and monomer, respectively. The joint antiparallel β-sheet containing four strands was formed in the dimer due to the interaction of the *N*-terminal β-strands of two capitellacin monomers (Figure 1B).

### 3.2. Spatial Structures of Monomer and Dimer of Capitellacin in DPC Micelles

The sets of the 20 spatial structures of the monomer and dimer of capitellacin (Figure 3A,B and Appendix A) were calculated using NMR data measured at 45 °C (Appendix A). Both forms of the peptide represent a β-hairpin formed by two antiparallel β-strands (Val4–Cys9 and Cys14–Trp19) enclosed by a β-turn (Arg10–Val13). Two disulfide bridges (Cys5–Cys18 and Cys9–Cys14) and eight intramonomer H-bonds stabilized the structures of these β-hairpins (Figure 1A,B). The β-sheets in the monomer and dimer have a right-handed twist of 53° ± 2° and 74° ± 6° (per six residues, ~9° and 12° per residue, Figure 3A,B), respectively. The comparison with the geometry of the capitellacin monomer in water (right-handed twist ~18°/res, Figure 3C [23]) revealed moderate flattening (untwisting) of the peptide β-sheet upon contact with detergent micelles. The incorporation of the peptide in micelles also changed the conformation of the Arg10–Val13 β-turn from type IV in water to type I’ for both forms of capitellacin in micelles. In the major forms of the capitellacin monomer and dimer present in solution (see Section 3.1), the Ser1–Pro2 peptide bond had a *trans*-configuration. The symmetric dimer assembly was stabilized by four pairs of intermonomer H-bonds (Arg3:HN–Cys9:CO, Cys5:HN–Arg7:CO, Arg7:HN–Cys5:CO, Cys9:HN–Arg3:CO, and symmetrical bonds from another monomer, Figure 1B).

An analysis of the distribution of the electrostatic potential and molecular hydrophobicity potential (*MHP*) on the surface of the capitellacin monomer (Figure 3D and Figure 4A) revealed its amphipathic properties. A wide hydrophobic band runs along the entire molecule, formed by the sidechains of Pro2, Val4, Ile6, Val8, Val13, Tyr15, and Trp19. A single charged sidechain Arg17 protrudes in the middle of this region. In contrast, the other side of the monomer contains all the other charged groups of the molecule: *N*-terminal amine, Arg3, Arg7, Arg10, Arg16, and *C*-terminal carboxyl (Figure 3D). In contrast, the capitellacin dimer demonstrates significantly increased amphipathicity (Figure 3E and Figure 4B). One side of the nearly planar dimer contains a large hydrophobic patch formed by the sidechains of both monomers. At the same time, all the charged groups of both monomers were located on the edge of the flat dimer and on its other side. Notably, the sidechains of the Cys residues were located on the polar side of the dimer.

### 3.3. Topology of the Capitellacin Interaction with DPC Micelles

The hydrophobic regions formed on the surfaces of the capitellacin monomer and dimer may be responsible for the favorable interaction of the peptide with lipid membranes and detergent micelles. To determine the position of the peptide in the micelle, the paramagnetic probe 12-doxylstearic acid was added to the samples of ^15^N-labeled capitellacin in predominately monomeric (P:D = 1:200) and dimeric (P:D = 1:50) forms. The paramagnetic group of the 12-DSA molecule was localized in the hydrophobic interior of the DPC micelle [50]. This group enhanced the relaxation and attenuated the amide proton signals of the *N*-terminal β-strand (Arg3–Cys9) in both peptide forms (Figure 5A). The signals of the β-turn (Arg10–Val13) and *C*-terminal β-strand (Cys12–Gly20) were less attenuated. The observed attenuation patterns were consistent with the surface binding of the peptide monomer and dimer to the micelle. Probably in both structural forms, capitellacin interacted with the micelle surface slightly immersing the *N*-terminal β-strand into the hydrophobic region (Figure 5B,C). Interestingly, the attenuation patterns at the *C*-termini of the monomer and dimer were different (Figure 5A). For example, the H^ε1^ signal of the Trp19 sidechain was strongly attenuated in the dimer, whereas it was of significant intensity in the monomer. This agrees with the greater twisting of the peptide β-sheet within the dimer compared to the monomer (see above). As a result, the sidechains of Trp19 in the dimer were turned toward the hydrophobic region of the micelle (Figure 5B). 

### 3.4. Thermodynamics of the Capitellacin Dimerization in DPC Micelles

The H^ε1^ proton of the Trp sidechain resonates in a separate region of the 1D ^1^H NMR spectrum. Capitellacin contains only one Trp19 residue, and this allows the population of the different peptide forms to be measured by approximating a region of the ^1^H 1D NMR spectrum using a sum of several Lorentzian lines. In each case, we observed an additional minor signal(s) with a population ~10% that probably corresponded to the peptide form(s) with a *cis*-configuration of the Ser1–Pro2 peptide bond (Figure 6A,C). At the initial points of the titration of the capitellacin sample in water with DPC, a transition of the peptide from the aqueous phase to the micelle-bound form was observed (Figure 6A). According to the observed position of the ^1^H^ε1^ Trp19 resonance, at 1.4 mM DPC (D:P = 10:1), capitellacin was in the aqueous phase, and increasing the DPC concentration to 4.2 mM (D:P = 30:1) led to the partial transition of the peptide into the micelle-bound form. The intermediate position of the ^1^H^ε1^ Trp19 resonance at D:P = 30:1 indicated fast (on the NMR timescale) exchange between the monomeric peptide in water and the dimeric peptide in micelles. The transition of capitellacin to the micelle-bound form was completed at 7.0 mM DPC (D:P = 50:1). Under these conditions, a single signal of the capitellacin dimer in micelles was observed. A further increase in the DPC concentration to 9.8 mM and above (D:P ≥ 70:1) led to the appearance of additional signals corresponding to the micelle-bound capitellacin monomer. The population of this form of the peptide increased with an increasing detergent concentration (Figure 6A). The exchange between the monomeric and dimeric capitellacin in micelles was slow (on the NMR timescale).

To estimate the thermodynamic parameters of the detergent-dependent capitellacin dimerization, we used the slightly modified ‘micellar solvent’ model (see Section 2.5, Equation (1)) [43]. This model provided a good approximation of the experimental data (Figure 6B), assuming that the average number of detergent molecules (aggregation number) in an empty (peptide-free) DPC micelle (N_e_) is 55. The fitting of the model resulted in the following parameters: the aggregation numbers of the micelles containing monomer (N_m_) or dimer (N_d_) of the peptide were 56 ± 1 and 59 ± 3, respectively; the apparent reaction order with respect to detergent micelles (γ) is about 0.96; and the free energy of the dimerization (Δ*G^D^*) was −3.0 ± 0.3 kcal/mol (at 30 °C). Almost equal aggregation numbers of DPC in the empty micelles and micelles with the capitellacin monomer and dimer (55–60 molecules) indicate that the peptide binding does not significantly perturb the micelle structure.

To obtain the thermodynamic parameters—the enthalpy (Δ*H^D^*) and entropy (Δ*S^D^*) of the capitellacin dimerization—Δ*G^D^* values were measured in the temperature range 5–50 °C with a D:P = 400:1 (Figure 6C,D). A decrease in temperature shifted the equilibrium toward dimer formation, which was observed by changes in the relative signal intensities in the spectra (Figure 6C). The observed linear temperature dependence of Δ*G^D^* (Figure 6D) revealed the absence of significant changes in heat capacity during dimerization (Δ*C_P_^D^*~0) and provided the following thermodynamic parameters: Δ*H^D^* = −8.88 ± 0.02 kcal/mol, *T*·Δ*S^D^* = −5.77 ± 0.10 kcal/mol (at 30 °C), and Δ*S^D^* = −19.0 ± 0.3 cal/mol/K. 

## 4. Discussion

### 4.1. Physicochemical Properties of β-Hairpin AMPs and Their Change upon Incorporation into a Membrane-like Environment and Dimerization

Several high-resolution structures of oligomeric AMPs have been determined via solution or solid-state NMR spectroscopy in membrane-like environments. In the environment of DPC micelles, two structures of oligomers of helical AMPs have been determined: an antiparallel symmetric dimer (↑↓) of a 22-residue analogue of amphibian magainin (MSI-78) [52] and an antiparallel/parallel symmetric tetramer (↑↓↑↓) of a 22-residue fragment of chicken cathelicidin fowlicidin-1 (VK22) [53]. Among β-hairpin peptides, antiparallel (NC↑↓CN, Figure 1F) dimers were described for protegrin-1 and protegrin-3 in DPC micelles [12,13], parallel (NC↑↑CN, Figure 1E) dimers of protegrin-1 were observed in POPC and POPG/POPE (3:1) bilayers [11,14], parallel (CN↑↑NC, Figure 1C) dimers were observed for arenicin-2 in DPC and SDS micelles and DPC/DOPG bicelles [15], an antiparallel (↑↓↑↓, Figure 1G) dimerization was described for ChDode (Cetartiodactyla cathelicidin-1) in DPC micelles [16], and an antiparallel (CN↑↓NC, Figure 1H) dimer was observed for thanatin in negatively charged LPS micelles [17]. 

We compared the physicochemical parameters and surface properties of the antiparallel capitellacin dimer (CN↑↓NC, Figure 1B) observed in the DPC micelle environment with those of other β-hairpin peptide dimers (Appendix A, Figure 4). To characterize the changes in β-hairpins induced by incorporation into a membrane mimetic and dimerization, we also analyzed the properties of 15 monomeric β-hairpin AMPs in various environments (see Appendix A, and Section 2.4). For the comparison, the structures of four monomeric helical AMPs (fowlicidin-1, melittin, magainin-2, and MSI-594, see Appendix A and Section 2.4) were also included. (The structures of oligomeric helical AMPs have not been deposited in the PDB database.)

Despite the different modes of dimerization, all β-hairpin AMPs demonstrated very similar overall properties, including molecular weight (2477 ± 304 kDa, mean ± SD, *n* = 15), amino acid sequence length (19.9 ± 2.4 residues), and charge (6.2 ± 1.3). At the same time, the average hydrophobicity of the peptide sequence calculated according to the Kyte and Doolittle hydrophobicity scale [25] (*GRAVY*) and content of aromatic residues (*AROM*) varied significantly (−0.24 ± 0.66 and 0.15 ± 0.09, respectively, Appendix A). The highest aromatic contents (≥0.24) were observed in arenicins and polyphemusins, and the highest *GRAVY* values (≥+1.0) were observed for ChDode and PcDode. The lowest *GRAVY* values (≤−0.8) are characteristic of thanatin, polyphemusins, and gomesin. A comparison of the surface properties for the monomeric β-hairpin peptides revealed that arenicin-2 and ChDode are significantly more hydrophobic than other AMPs (Appendix A). The average area of the highly hydrophobic surfaces (*S_lip_*) for all monomeric β-hairpins was 432 ± 128 Å^2^ (mean ± SD, *n* = 19), while for arenicin-2 and ChDode *S_lip_*, it was 673 and 663 Å^2^, respectively. For these two peptides, the mean *MHP* value on the surface (*MHP_mean_*~+0.04) and relative area of the hydrophobic surface (*S_lip_*/*S_tot_*~0.34) were also significantly higher than the average values (−0.09 ± 0.09 and 0.25 ± 0.06, respectively). Surprisingly, in the case of arenicin-2, a high surface hydrophobicity does not correlate with the *GRAVY* value (−0.06, Appendix A). On the other hand, the monomers of protegrin-1, protegrin-2, alvinellacin, capitellacin, thanatin, and gomesin in water exhibited the lowest surface hydrophobicity values (*S_lip_* ≤ 320 Å^2^, *S_lip_*/*S_tot_* ≤ 0.20, and *MHP_mean_* ≤ −0.09).

The incorporation of monomeric β-hairpin AMPs into a membrane mimetic (DPC micelles) does not significantly change the surface hydrophobicity (Appendix A, orange vs. gray bars), while dimerization in a membrane-like environment results in an approximately twofold increase in hydrophobic surface area (*S_lip_*) for all peptides except thanatin (Appendix A, blue vs. orange and gray bars). The parameters describing the relative surface hydrophobicity (*S_lip_*/*S_tot_* and *MHP_mean_*) also show an increase upon dimerization (Appendix A).

Thus, the β-hairpin AMPs dimerize in the membrane-like environment through the association of polar or mildly hydrophobic surfaces, keeping highly hydrophobic regions available for interaction with the membrane. This is contrary to the situation observed for the oligomers of helical peptides MSI-78 and VK22, which oligomerize through the contact of hydrophobic surfaces [52,53]. Interestingly, the synthetic magainin/melittin hybrid MSI-594 in LPS micelles forms a monomeric α-helical hairpin with a deep kink in the middle, so the hydrophobic residues of the *N*-terminal and *C*-terminal helices are in contact with each other and are partially hidden from contact with the environment [38]. Similarly, the β-hairpin peptide thanatin also oligomerizes in LPS micelles due to hydrophobic surface contact [17] (Appendix A, *S_lip_* in the two thanatin monomers: ~300 × 2 = 600 Å^2^, while *S_lip_* in the dimer: ~300 Å^2^). It should be noted that thanatin is not a typical membrane-active AMP since it does not directly act on the lipid bilayer but on specific proteins of Gram-negative bacteria that are responsible for the stability of the outer membrane and LPS transport [54,55]. 

### 4.2. Hydrophobicity of the β-Hairpin AMPs Correlates with the Stability of Dimers in a Membrane-like Environment

The discrepancy between the *GRAVY* values and surface hydrophobicity indicates that the hydropathy scale used does not correctly reproduce the properties of β-hairpin peptides. We therefore optimized a linear combination of the two parameters *GRAVY* and *AROM* against different parameters describing overall surface hydrophobicity (*S_lip_*, *S_lip_*/*S_tot_*, *MHP_mean_*, and *MHP_sum_*) over a set of monomeric β-hairpin AMPs. The best correlation (Pearson’s squared correlation coefficient R^2^ = 0.76, *p* < 0.0001, *n* = 15) was observed between the *S_lip_*/*S_tot_* parameter and the *GRAVY* + 8 × *AROM* function (Figure 7A). Other parameters yielded R^2^ values in the range 0.30 ÷ 0.65 (*p* = 0.03 ÷ 0.0003). Interestingly, this analysis showed that the surface properties of helical AMPs can differ significantly from those of β-hairpin peptides; the corresponding points lay in a separate region of the graph, deviating from the proposed linear trend (Figure 7A). (Note the very limited set of helical peptides used). On the other hand, the data points describing the dimers of β-hairpin peptides, with the exception of arenicin-2, followed a linear trend and were within the range of the *S_lip_*/*S_tot_* values observed for monomeric peptides (Figure 7A). Thus, changes in the relative hydrophobic surface area upon dimerization are within the natural variation in this parameter in monomeric β-hairpin peptides. In contrast, arenicin-2 is characterized by a very large increase in hydrophobic surface area upon dimerization.

The observed difference in the surface hydrophobicity correlates with the relative stability of the peptide dimers. Indeed, arenicin-2 and ChDode in the DPC environment demonstrated only dimeric structures (and possibly higher-order aggregates), that did not dissociate into monomers when the detergent content in the samples was increased to D:P ratios of 130–200:1. At the same time, all previous studies on protegrins using solution or solid-state NMR [11,12,13,14] were conducted at fairly low D:P or Lipid:Peptide (L:P) ratios (≤20:1). A solid-state ^19^F NMR study on protegrin-1 in POPC bilayers revealed that at a L:P ratio of 35:1 the dimer fraction was only 60%, and it increased to 88% at a L:P of 12.5:1 [56]. These data were qualitatively comparable to the results of the current capitellacin study (Figure 6A), where ~73% of the dimer was observed at a D:P of 70:1 (note that correction for the critical micelle concentration results in a real D:P ratio in the micelle of ~60:1). A comparison of the free energy of dimerization (Δ*G^D^*) also revealed good correspondence between the properties of protegrin-1 and capitellacin. For protegrin-1 in the POPC bilayer, a Δ*G^D^* of −10.2 ± 2.3 kJ/mol (−2.4 ± 0.5 kcal/mol) was reported [56], which is similar to the −3.0 ± 0.3 kcal/mol value observed for capitellacin. 

The strong dependence of β-hairpin dimerization on peptide hydrophobicity was recently demonstrated in a study on the [Val8/Arg] mutant of arenicin-1 [21]. Arenincin-1 forms dimers that are stable in DPC micelles at least up to a D:P of 200:1, while the replacement of one hydrophobic residue with a charged one (Val8/Arg) leads to a weakly dimerizing peptide, which, in DPC solution, exhibits an exchange between several structural states, the major of which is a micelle-bound monomer [21]. Another example is one of the least hydrophobic peptides in our set, thanatin. This β-hairpin molecule does not dimerize in zwitterionic DPC micelles [22], and only acquires a dimeric structure in anionic LPS micelles, where additional electrostatic interactions between the cationic peptide and anionic lipid can support dimerization [17]. Interestingly, the strong dependence of peptide dimerization or oligomerization on hydrophobicity was also shown for helical AMPs [57].

The color-code used for the different β-hairpin peptides in Figure 7A permits the visualization of the relative propensity of the peptides to form dimers in the DPC micelles environment. It can be seen that the peptides that form stable dimers (arenicin-2 and ChDode, black circles) and the peptides that form weak dimers (capitellacin, protegrin-1, and protegrin-3, blue circles) are grouped in two non-overlapping regions of the graph and differ by surface hydrophobicity (*S_lip_*/*S_tot_* parameter) and amino acid sequence hydrophobicity (*GRAVY* + 8 × *AROM* function). Using this graph, we can predict that arenicin-3 should form stable dimers even in a zwitterionic environment. Arenicin-1 [Val8/Arg], polyphemusin, PcDode, PV5, tachyplesin, and protegrin-2 can form dimers in anionic membrane mimetics or in zwitterionic media at very low D:P or L:P ratios; while alvinellacin and gomesin probably do not form dimers at all, or can form them in anionic lipids. 

For many of the β-hairpin peptides studied, dimers have not previously been detected in a solution of DPC micelles (Figure 7A, light blue circles); this is likely a consequence of the weak binding of the peptides to micelles at low D:P ratios. In this case, the probable dimerization of the membrane-bound peptide is obscured by the high concentration of the monomeric peptide present in the aqueous phase. On the other hand, an increase in the D:P ratio simultaneously leads to an increase in the micelle binding and to the destabilization of dimers. Thus, the observed correlation between the hydrophobicity of β-hairpin AMPs and dimer stability can be partly explained by the greater membrane affinity for the more hydrophobic peptides. Moreover, the addition of anionic lipids to the membrane mimetic can increase the binding of cationic peptides and hence increase their dimerization.

### 4.3. The Hydrophobicity and Total Charge of β-Hairpin AMPs, but Not the Ability to Form Dimers, Are Positively Correlated with Hemolytic Activity

The knowledge of spatial structures for the set of β-hairpin AMPs makes it tempting to search for correlation between their physicochemical properties and activity. The antimicrobial action of some β-hairpin AMPs is based on specific mechanisms (e.g., tachyplesin targets LPS [58], thanatin attacks the periplasmic proteins that transport LPS molecules [54], etc.). The involvement of these specific mechanisms complicates the search for such a correlation with antimicrobial activity. At the same time, the interaction of AMPs with eukaryotic cells probably reflects the nonspecific membrane activity of the peptides, which may indeed correlate with charge, hydrophobicity, and other properties. The most convenient for analysis is hemolytic activity, since it has been measured for many peptides.

Previous investigations of helical AMPs have shown that increases in the hydrophobicity, amphipathicity, and rigidity of the helical structure increase hemolytic activity [57,59]. On the other hand, a comparative study on several β-hairpin AMPs showed that the greatest toxicity for eukaryotic cells was observed in peptides with high amphipathicity and moderate hydrophobicity [39]. Therefore, we tested the possible correlation between the hydrophobicity and charge of β-hairpin AMPs and their hemolytic activity. For the analysis, we took the minimal hemolytic concentration (*MHC*) or the concentration at which 10% hemolysis was observed. A comparison of different literature sources yielded *MHC* values for 12 β-hairpins and four helical AMPs from our dataset (see Section 2.4). The *CHARGE* parameter and the *GRAVY* + 8 × *AROM* function showed weak negative non-significant correlations with the lg(*MHC*) values (R^2^ = 0.14 and 0.19, *p* = 0.24 and 0.16, *n* = 12, respectively). At the same time, the combination of these parameters (*GRAVY* + 8 × *AROM* + 0.5×*CHARGE*) provided a significant moderately negative correlation (R^2^ = 0.35, *p* = 0.043, *n* = 12, Figure 7B, blue line). The inclusion of the helical peptides in the analysis reduced the magnitude and significance of the correlation (R^2^ = 0.15, *p* = 0.14, *n* = 16, Figure 7B, red line).

The combination of parameters that describe surface hydrophobicity with charge also provided significant negative correlations with hemolytic activity. For example, the function *MHP_mean_* + 0.06 × *CHARGE* correlates with the lg(*MHC*) values for monomeric β-hairpin AMPs with R^2^ = 0.36 (*p* = 0.039, *n* = 12, Figure 7C, blue line). The inclusion of the β-hairpin dimers and helical peptides in the analysis provided a more significant, but still moderate, correlation (R^2^ = 0.42, *p* = 0.0014, *n* = 21, Figure 7C, red line). Interestingly, the dimeric β-hairpin peptides did not show an alternative dependence (Figure 7C). As noted above, the change in surface hydrophobicity during peptide dimerization is within the spread of values observed for monomeric β-hairpins. 

The dependence of AMP toxicity on the peptide hydrophobicity is consistent with the results of previous studies [39,57,59]. In our dataset, this relationship can be illustrated by a pair of homologues peptides: arenicin-2 and arenicin-1[Val8/Arg]. The first peptide is more hydrophobic (*MHP_mean_* = 0.04 vs. −0.23, *S_lip_* = 673 vs. 545 Å^2^, Appendix A) and, at the same time, much more hemolytic (*MHC* = 4 vs. 125 μM). On the other hand, the positive correlation between the total peptide charge and hemolysis appears counterintuitive. However, our dataset provides several examples of such dependence. In the pair of homologues peptides tachyplesin and capitellacin, the monomeric peptides in the DPC micelles environment show a similar surface hydrophobicity (*MHP_mean_* = −0.05 vs. −0.07, *S_lip_* = 415 vs. 409 Å^2^, Appendix A), but tachyplesin has a higher charge (+7 vs. +5) and is much more hemolytic (*MHC* = 25 vs. >128 μM). Similarly, polyphemusin and arenicin-3 have a very similar surface hydrophobicity in water (*MHP_mean_* = −0.01 vs. +0.01, *S_lip_* = 505 vs. 529 Å^2^, Appendix A), but polyphemusin has a significantly higher charge (+8 vs. +4) and exhibits increased hemolytic activity (*MHC* = 25 vs. 130 μM). 

The color code used for the different β-hairpin AMPs in Figure 7B,C reveals no correlation between the ability to form dimers and hemolytic activity. Previously, the opposite was assumed for helical peptides. It has been suggested that large dimers or oligomers poorly penetrate the dense layers of LPS or peptidoglycan on the surface of bacterial cells, but, at the same time, they are good at the nonspecific lysis of eukaryotic membranes [57]. However, in Figure 7B,C we see that the peptides that dimerize in DPC micelles (black and blue circles) occupy a wide range on the *MHC* scale (from 4 to >128 μM). On the other hand, the presence of cross-correlations between hydrophobicity and dimer stability, as well as between hydrophobicity and hemolytic activity, suggests that a relationship between hemolysis and tendency to dimerize is possible. 

It should be noted that in our analysis, we did not take into account such an important parameter as amphipathicity. This parameter can also change dramatically during peptide dimerization. The literature shows examples of a strong dependence of the antimicrobial activity and of the toxicity of AMPs on amphipathicity [39,59].

### 4.4. The Distribution of Hydrophobic and Polar Regions on the Capitellacin Surface Determines the Surface Mode of the Peptide/Micelle Interaction and the Carpet Mechanism of Action on Membranes

Typically, the β-hairpins of AMPs exhibit significant right-handed twist and bending in aqueous solution [26]. These deformations effectively shield the hydrophobic regions on the surface of the peptides from contact with water. However, interaction with micelles and dimerization leads to the flattening (untwisting) of the peptide β-structure, accompanied by a change in the β-turn conformation and the formation of nearly planar amphipathic dimers with two large faces. Previously, such conformational rearrangements were observed for arenicin-2 and ChDode [15,16], and now we have observed them for capitellacin as well (Figure 3B,C). The distribution of the molecular hydrophobicity potential on the surfaces of known dimers of β-structural AMPs (Figure 4) showed that both faces of planar dimers contain large hydrophobic regions, while charged and polar groups are localized mainly at the edges. This is not the case for the capitellacin dimer, which has only one hydrophobic face, while the other face is polar and accommodates two pairs of positively charged groups belonging to the Arg7 and Arg16 residues of both monomers (Figure 3E and Figure 4B). 

The observed difference in the distribution of hydrophobic regions on the surface results in the different modes of peptide/micelle and peptide/membrane interactions and likely determines the primary mechanism of the membrane activity of the peptide. The presence of two hydrophobic regions on the surfaces of the arenicin-2 and ChDode dimers, as well as the localization of most charged groups at the *N*- and *C*-termini and in the β-turn region, leads to the ‘transmembrane’ (TM) position of the peptides in DPC micelles [15,16]. Similarly, the incorporation of the protegrin-1 dimer into the hydrophobic interior of the POPG/POPE bilayer was observed [11]. The TM topology is consistent with the different mechanisms of pore formation [60], which may explain the effects of the peptides on bacterial membranes. For arenicin, ChDode, and protegrin-1, large ion-conducting ‘toroidal’ pores were formed by several peptide dimers, and lipid molecules have been proposed [11,15,16], while octameric or decameric ‘barrel-stave’ pores formed exclusively from the peptide dimers (β-barrels) have been suggested for protegrin-3 [13].

At the same time, the data obtained here for the monomer and dimer of capitellacin are consistent only with their binding to the surface of DPC micelles (Figure 5). Interestingly, the *N*-terminal strand (Arg3–Cys9) of the β-hairpin, which is used for the peptide dimerization, is more deeply immersed in the micelle, compared to the rest of the molecule. This results in an overall boat-like shape for the dimer (Figure 5B). Probably, the most energetically favorable state of the capitellacin dimer in the membrane is the surface-bound state, in which the peptide is located in the lipid headgroup region. The presence of bulky charged sidechains in the interior of the ‘boat’ does not allow it to incorporate into the hydrophobic region of the membrane in a TM manner. This topology better corresponds to the ‘carpet’ mechanism of membrane activity [60]. We hypothesize that capitellacin forms dimers and higher-order aggregates on the surface of bacterial membranes. The accumulation of the peptide destabilizes the bilayer and disrupts its integrity due to micellization without the formation of quasi-stable TM pores. This may be the primary mechanism of capitellacin’s antibiotic action, consistent with previous biochemical and electrophysiological studies [6,23]. 

Capitellacin demonstrates significantly lower antibacterial, hemolytic, and membrane permeabilizing activity compared to the homologous horseshoe crab peptide tachyplesin-1 [6,23] (Figure 1C). Mutagenesis has shown that the key parameter responsible for this difference is the amino acid sequence of the β-turn (Arg10–Asn–Gly–Val13 in capitellacin and Tyr8–Arg–Gly–Ile11 in tachyplesin) [6]. A substitution of the Arg–Asn dipeptide fragment with Tyr–Arg significantly increases the activity of the capitellacin analogue. The β-turn regions form the bow and stern of the ‘boat’ of the capitellacin dimer, according to our data (Figure 5B), are localized in the aqueous phase. The positively charged Arg10 groups protruding on the hydrophobic face of the capitellacin dimer (Figure 3E and Figure 4B) probably prevent the immersion of the β-turn regions into the membrane (Figure 5B). The Arg10/Tyr substitution in tachyplesin increases the hydrophobicity of the β-turn and allows the peptide to integrate deeper into the hydrophobic region of the bilayer. The Asn11/Arg substitution probably does not interfere with this, since the Asn11 sidechains do not protrude onto the hydrophobic face of the capitellacin dimer (Figure 5B). Indeed, in contrast to the carpet mechanism proposed here for capitellacin, the formation of toroidal pores in anionic membranes has been described for tachyplesin [61,62]. Differences in the mechanisms explain the slow kinetics and relatively low activity of membrane permeabilization by capitellacin [6]. 

Based on the available structures of dimeric β-hairpin AMPs (Figure 4), we can assume that peptides forming planar dimers (with the exception of capitellacin) are able to form TM pores in the bilayer, whereas monomeric peptides and capitellacin tend to act through the carpet mechanism. Presumably, the disruption of the membrane barrier by TM pores can be achieved at a lower peptide concentration than bilayer micellization with carpet-like oligomers. Thus, the first type of mechanisms should generally be more efficient, and the dimeric peptides should be more active than the monomeric ones. At the same time, the data presented in Figure 7A predict that virtually all β-hairpin AMPs can dimerize (with varying efficiencies) in bacterial anionic membranes. Thus, a particular peptide can act on the membrane through a variety of possible mechanisms, depending on the state of the dimer-monomer equilibrium under specific conditions. In this case, identifying the single mechanism responsible for the antimicrobial activity may be challenging. Likewise, measuring activity separately for the monomeric and dimeric forms of AMPs is also difficult. However, the temperature dependence of capitellacin dimerization (Figure 6) suggests a possible way to achieve this by measuring membrane and/or antimicrobial activity at different temperatures. Such experiments are complex and should be planned for future research.

It should be noted that the discussion above can be valid only for β-hairpin peptides. Helical AMPs can form TM pores by assembling individual monomers. In addition, helical peptides, which act on the membrane exclusively through the carpet mechanism, sometimes demonstrate quite high efficiencies [63].

### 4.5. The Dimerization of Capitellacin in DPC Micelles Is an Exothermic- and Enthalpy-Driven Process

The slow exchange between the micelle-bound monomer and dimer of capitellacin allowed us to estimate the thermodynamic parameters of the peptide dimerization in the detergent–protein–water system (Figure 6). Two main features could be noted. First, the dimerization is an exothermic process: the dimer becomes less stable and reversibly dissociates to monomers at high temperatures (Figure 6C,D). Second, the dimerization is an enthalpy-driven process (Δ*H^D^* < 0 and Δ*S^D^* < 0), which means that the peptide self-association depends more on intermolecular interactions (hydrogen bonds, electrostatic, van der Waals) than on the hydrophobic effect. Indeed, the close similarity of the peptide structure in monomeric and dimeric forms (Figure 3A,B), the similarity of the properties of detergent micelles with the monomer and dimer incorporated (Nm ≈ Nd), and the similarity of the interactions of the monomer and dimer with the micelle surface (Figure 5) suggests that the observed free energy change Δ*G^D^* is mainly due to the formation of intermolecular hydrogen bonds and van der Waals contacts, which clearly contribute to the negative enthalpy. Using a model AcWL_5_ peptide, White and co-workers estimated the free energy reduction for β-sheet formation in the membrane to be ~0.5 kcal/mol per residue (per hydrogen bond) [64]. Thus, the Δ*G^D^* value determined for capitellacin (~−3.0 kcal/mol) approximately corresponds to the formation of six intermonomer hydrogen bonds in a membrane-like environment, which is consistent with the observed changes in ∆δ^1^H^N^/∆T (Figure 2C) and the proposed dimer structure (Figure 1B). 

To our knowledge, there have been no previous studies describing the thermodynamic parameters of the dimerization of β-structural peptides in membranes. However, we can compare the stability of the capitellacin dimer with those of β-barrel membrane proteins. These proteins exhibit a wide range of free energy changes upon reversible unfolding from the membrane-inserted state, ranging from ca. 9 to 30 kcal/mol [65]. These changes correspond to an increase in free energy of 0.08 ÷ 0.25 kcal/mol per one residue inserted into the membrane and are compatible with the Δ*G^D^* value of capitellacin, which has 18 residues at the dimerization interface (Figure 1B). 

The Δ*G^D^* value of capitellacin can also be compared with the reported dimerization energies of the TM α-helical domains of the membrane protein glycophorin A (Δ*G^D^* from −4 to −12 kcal/mol depending on membrane/detergent properties [45,47,66,67]) and the receptors of the RTK family (Δ*G^D^* from −1.4 to −3.4 kcal/mol [43,68]). These data show a comparable stability of the capitellacin dimer, even though the helical domains occupy the TM position in the micelles, and capitellacin is bound to the surface.

The linear temperature dependence of the dimerization free energy Δ*G^D^* (Figure 6D) indicates that there was no change in the heat capacity during capitellacin dimerization (Δ*C_P_^D^*~0). Generally, heat capacity is related to the number of degrees of freedom of the system. When unfolding water-soluble globular proteins, the heat capacity change Δ*C_P_^U^* is positive and dominated by the change in the hydration of the hydrophobic and polar residues. So, the Δ*C_P_^U^* value becomes proportional to the increase in solvent accessible surface area (SASA) during protein unfolding and to the number of water molecules that additionally bind to the unfolded protein as compared to a folded one [69,70]. In contrast, the unfolding of membrane proteins is additionally accompanied by changes in the lipid (detergent) organization around the protein, so the available data are very contradictory. For example, zero heat capacity changes were reported for the dimerization of the glycophorin TM domain in LDAO and SDS micelles [45]; a significant negative Δ*C_P_^U^* value was observed upon the dissociation of the dimer of the TM α-helical domain of the ErbB4 receptor in DMPC/DHPC bicelles [68]; a positive Δ*C_P_^U^* value was reported for the unfolding of the β-barrel membrane domain of AIDA protein in mixed Octyl-POE/SDS micelles [71]. In the case of capitellacin, it can be noted that the monomers dimerize through the association of *N*-terminal β-strands already immersed in the micelle, and the size and packing of the DPC micelle are probably not changed during dimerization. Thus, neither the peptide–water interactions nor the peptide–detergent interactions are significantly altered by dimerization, which explains the lack of detectable changes in heat capacity. 

## 5. Conclusions

It is generally accepted that the dimerization of membrane-active AMPs increases the efficiency of their interaction with bacterial membranes and, accordingly, leads to an increase in antimicrobial activity [4]. In this study, we found that the β-hairpin AMP capitellacin from marine polychaeta dimerize in a membrane-mimicking micellar environment through the antiparallel association of *N*-terminal β-strands (CN↑↓NC). Both the monomer and dimer of the peptide bound to DPC micelles were observed simultaneously, allowing a detailed structural and thermodynamic study via NMR spectroscopy. The process of capitellacin dimerization in micelles is enthalpy driven, characterized by negative changes in enthalpy and entropy, possibly corresponding to the process occurring under native conditions in bacterial membranes. As the temperature decreases, the equilibrium of the dimerization reaction shifts toward the dimer formation. The optimal habitat temperature for the marine polychaeta *Capitella teleta* is less than 15 °C. According to our data, in this temperature range, the peptide should be predominantly in the dimeric state; this state of the peptide likely has greater membrane activity and is responsible for the antimicrobial activity in vivo. Unlike other β-hairpin AMPs that form pores in target membranes, the distribution of the polar and hydrophobic regions on the surface and interface-bound topology of the capitellacin/membrane interaction is consistent only with the ’carpet’ mechanism of membrane activity. This may be the primary mechanism of capitellacin antibiotic action, which explains its lower antibacterial, hemolytic, and membrane permeabilizing activities compared to the β-hairpin peptides from other organisms. In addition, a comparison of the structural and physicochemical properties of capitellacin and other known β-hairpin AMPs in monomeric and dimeric forms showed the strong correlations of the dimer stability and hemolytic activity with the AMPs’ hydrophobicity.

To the best of our knowledge, this work represents the first detailed study on the thermodynamics of the dimerization of a β-hairpin peptide in a membrane-like environment. The results will allow us to more accurately describe the mechanisms of folding of large β-barrel membrane proteins and the formation of oligomeric transmembrane pores through β-structural AMPs and membranolytic toxins. We believe that studying the relationship between the spatial structures of antimicrobial peptides and their biological activities will help to design new peptide-based antibiotic drugs with high antimicrobial activities.

## Figures and Tables

**Figure 1 biomolecules-14-00332-f001:**
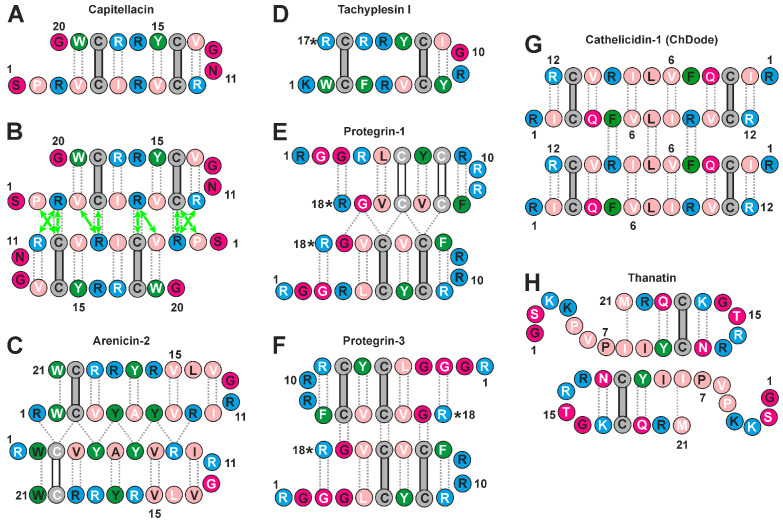
Secondary structures of capitellacin and other β-hairpin antimicrobial peptides in membrane-mimicking environments. (**A**,**B**) Monomer and antiparallel (symmetric) dimers of capitellacin observed in DPC micelles (present work). (**C**) Parallel (asymmetric) dimer of arenicin-2 observed in DPC micelles [15]. (**D**) Monomer of tachyplesin I observed in DPC micelles [18]. (**E**) Parallel (asymmetric) dimer of protegrin-1 observed in POPC and POPE/POPG (3:1) bilayers [11,14]. (**F**) Antiparallel (symmetric) dimer of protegrin-3 observed in DPC micelles [13]. (**G**) Antiparallel (symmetric) non-covalent dimer of the antiparallel disulfide-linked homodimers of cathelicidin-1 (ChDode, tetramer) observed in DPC micelles [16]. (**H**) Antiparallel (symmetric) dimer of thanatin observed in lipopolysaccharide (LPS) micelles [17]. Aromatic, hydrophobic, polar, positively charged, and Cys residues are shown as green, pink, magenta, blue, and grey circles, respectively. Asterisks indicate *C*-terminal amides. Disulfide bonds and H-bonds are shown as bars and gray dotted lines, respectively. Residues with sidechains directed up from the picture plane (toward readers) are marked with black labels; the residues with sidechains facing down from the picture plane are marked with white labels. The green arrows on panel (**B**) show intermonomer NOE contacts observed for the capitellacin dimer.

**Figure 2 biomolecules-14-00332-f002:**
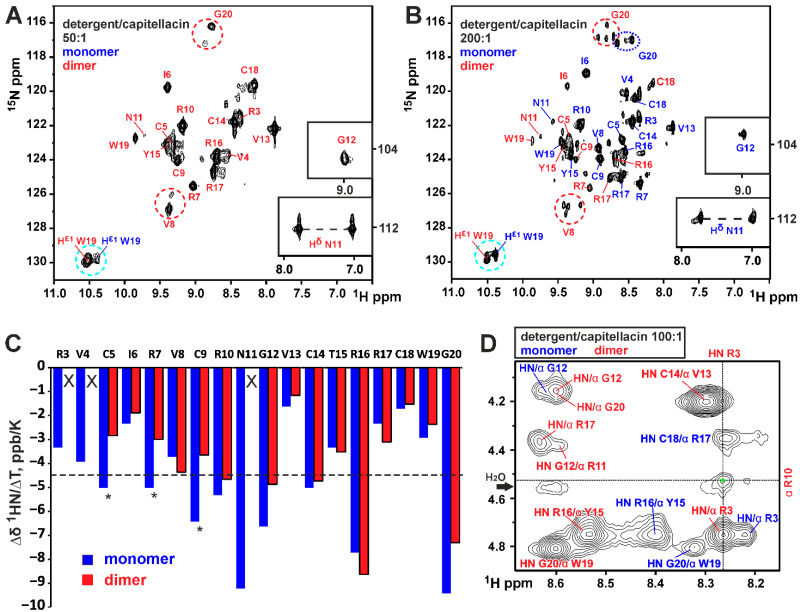
NMR data on capitellacin dimerization in DPC micelles. (**A**,**B**) 2D sensitivity-enhanced ^15^N-HSQC spectra of ^15^N-labeled capitellacin at D:P of 50:1 (**A**) and 200:1 (**B**) (0.07 mM, pH 5.4, 45 °C). Resonance assignments for monomer (M) and dimer (**D**) are shown in blue and red colors, respectively. The Val8 and Gly20 signal sets, probably arising from the *cis–trans* isomerization of the Ser1–Pro2 peptide bond in the monomer and dimer of capitellacin, are highlighted by blue and red circles. The H^ε1^ signals of the Trp19 sidechains in the monomer and dimer are highlighted by cyan circles. (**C**) Temperature gradients of amide protons (∆δ^1^H^N^/∆T) for monomer (blue columns) and dimer (red columns) measured at D:P of 200:1 and 50:1, respectively. The values > −4.5 ppb/K (dashed line) indicate the possible participation of the HN group in hydrogen bond formation. The H^N^ signals of Arg3, Val4, and Asn11 in the dimer (marked by crosses) were not observed at temperatures below 40 °C due to line broadening. Asterisks indicate HN groups that did not form H-bonds in the monomer but formed H-bonds in the dimer. (**D**) The fragment of a 2D NOESY spectrum of capitellacin at a D:P of 100:1 (0.07 mM, pH 5.4, 45 °C). The intermonomer Arg3:HN–Arg10:HA NOE-contact is indicated by green dot. The corresponding contacts are shown by arrows in Figure 1B.

**Figure 3 biomolecules-14-00332-f003:**
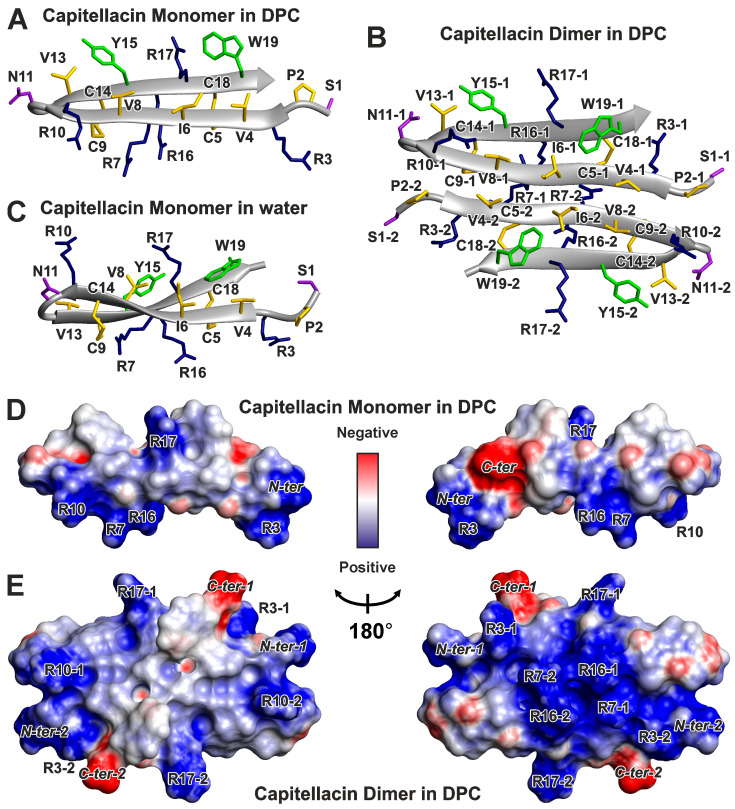
(**A**–**C**) Representative conformers of the capitellacin monomer and dimer in DPC micelles (present work, PDB ID: 8B4R and 8B4S, respectively) and capitellacin monomer in water (PDB ID 7ALD [23]). Residues from each monomer of the dimer are marked by “1” or “2”. Positively charged, aliphatic (including Cys), aromatic, and polar residues are colored blue, yellow, green, and magenta, respectively. (**D**,**E**) Two-sided view of electrostatic potential on the molecular surface of the capitellacin monomer and dimer in DPC micelles. Red and blue denote negative and positive regions, respectively.

**Figure 4 biomolecules-14-00332-f004:**
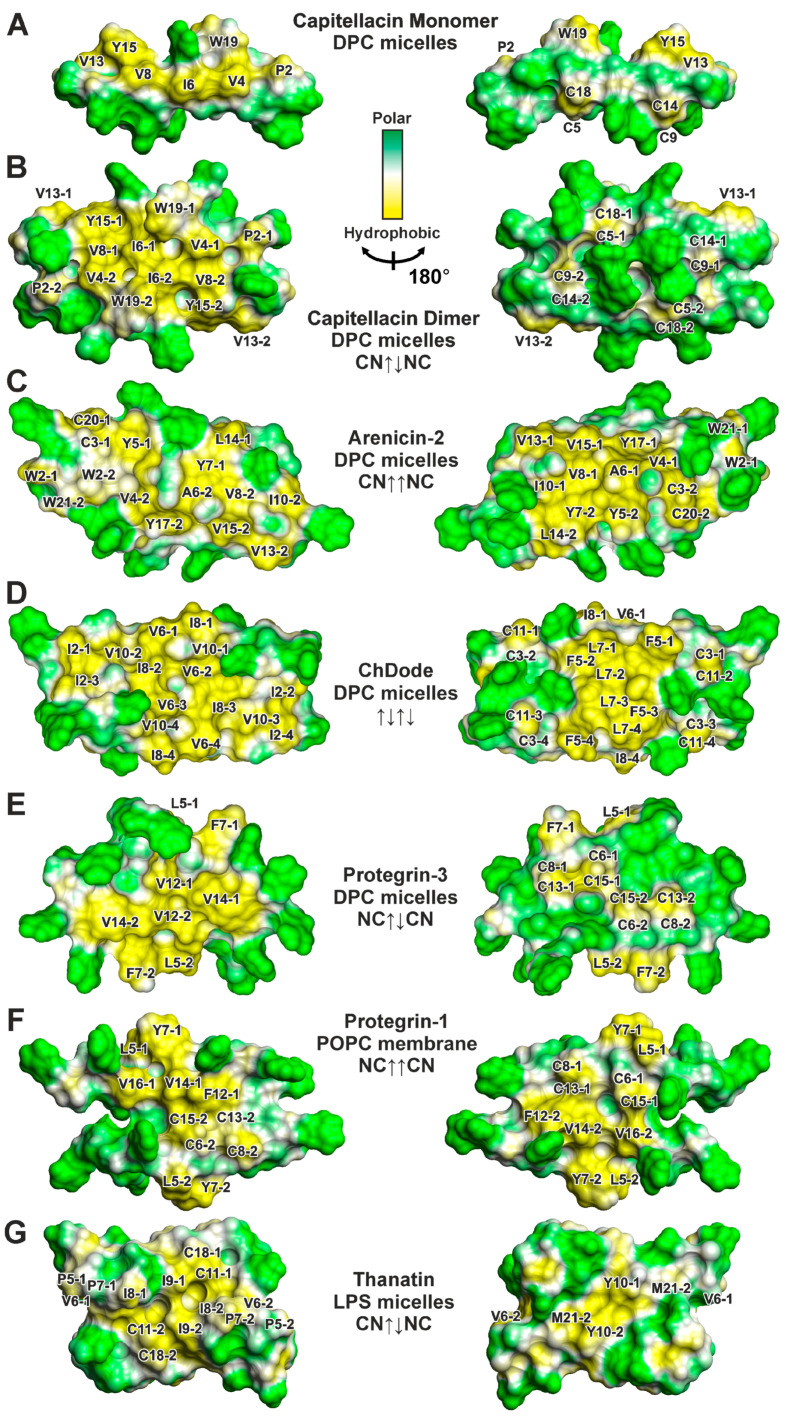
Two-sided views of molecular hydrophobicity potential (*MHP*) [27] on the molecular surface of the (**A**) capitellacin monomer (PDB ID 8B4R, present work), (**B**) capitellacin symmetric dimer (PDB ID 8B4S, present work), (**C**) asymmetric dimer of arenicin-2 (PDB ID 2L8X [15]), (**D**) symmetric tetramer of cathelicidin-1 ChDode (PDB ID 7ACB [16]), (**E**) symmetric dimer of protegrin-3 (PDB ID 2MZ6 [13]), (**F**) asymmetric dimer of protegrin-1 (PDB ID 1ZY6 [14]), and (**G**) symmetric dimer of thanatin (PDB ID 5XO9 [17]). Different molecules in the dimers and tetramer are denoted by numbers (‘1’, ‘2’, ‘3’, and ‘4’). Green and yellow denote polar and hydrophobic regions, respectively. Please note that two sides of symmetric dimers are different and contain different sidechains, while two sides of asymmetric dimers contain the same sidechains, and are similar, but not identical.

**Figure 5 biomolecules-14-00332-f005:**
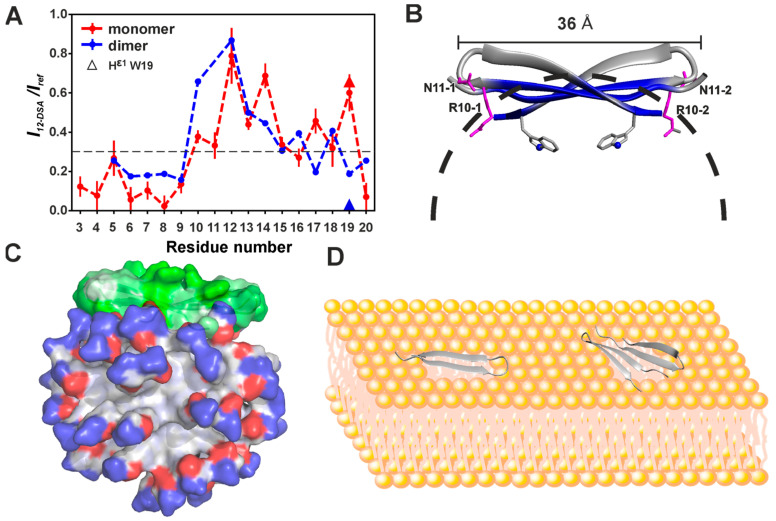
(**A**) Attenuation of the HN cross-peaks in the 2D ^15^N-HSQC spectrum of the capitellacin monomer (red) and dimer (blue) through a lipid-soluble paramagnetic probe 12-doxylstearate. Values for the H^ε1^ signal of Trp19 are shown as red and blue triangles. The 0.3 threshold line separates the residues in contact with the hydrophobic region of the micelle (below the threshold). (**B**) A model of the capitellacin dimer/micelle complex. The regions of the peptide affected by 12-DSA are shown in blue. Residues Arg10 and Asn11 are colored magenta. Different molecules in the dimer are denoted by numbers (‘1’, ‘2’). The dashed line represents the micelle surface with radius 21 Å. (**C**) Proposed model of the capitellacin dimer/DPC micelle complex. The peptide molecular surface is colored by the distribution of the molecular hydrophobicity potential. Green represents the polar surface area, and hydrophobic regions are colored white. The structure of a micelle containing 60 detergent molecules was generated in CHARMM-GUI [51]. The positively charged choline groups and negatively charged phosphate groups of the DPC molecules are colored blue and red, respectively. (**D**) Hypothetical model of the interaction of the capitellacin monomer and dimer with a bilayer.

**Figure 6 biomolecules-14-00332-f006:**
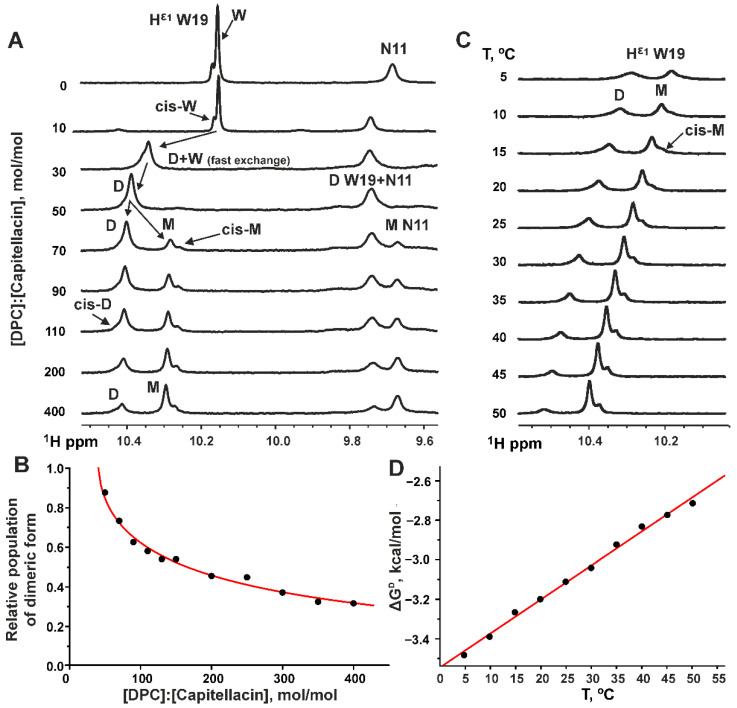
Thermodynamics of capitellacin dimerization in DPC micelles. (**A**,**B**) Titration of a 0.14 mM unlabeled capitellacin sample with DPC (pH 5.4, 30 °C). The peptide in water and in DPC micelles demonstrates additional conformational heterogeneity due to the *cis-trans* isomerization of the Ser1–Pro2 peptide bond. W, D, and M are the signals of the peptide monomer in water, as well as the capitellacin dimer and monomer in DPC micelles with the *trans*-configuration of the Ser1–Pro2 peptide bond. cis–W, cis–M, and cis–D are the signals of minor forms with a *cis*–Ser1–Pro2 bond. The D:P ratios shown in panel A were not adjusted for the presence of non-micellar detergent in the sample (the critical micelle concentration of DPC is 1.5 mM). The data in panel **B** were fitted using the ‘micellar solvent’ model [43]. See the fitting parameters in Section 3.4 of the manuscript. (**C**,**D**) Temperature dependence of the capitellacin NMR spectrum and linear approximation of the temperature dependence of the free energy (ΔG^D^) of capitellacin dimerization in DPC micelles (0.14 mM, D:P = 400:1, pH 5.4). The ΔG^D^ values were calculated from the measured concentrations of the monomeric and dimeric capitellacin using the ‘micellar solvent’ model with fitted parameters. The obtained values of ΔH^D^ and ΔS^D^ are provided in Section 3.4 of the manuscript.

**Figure 7 biomolecules-14-00332-f007:**
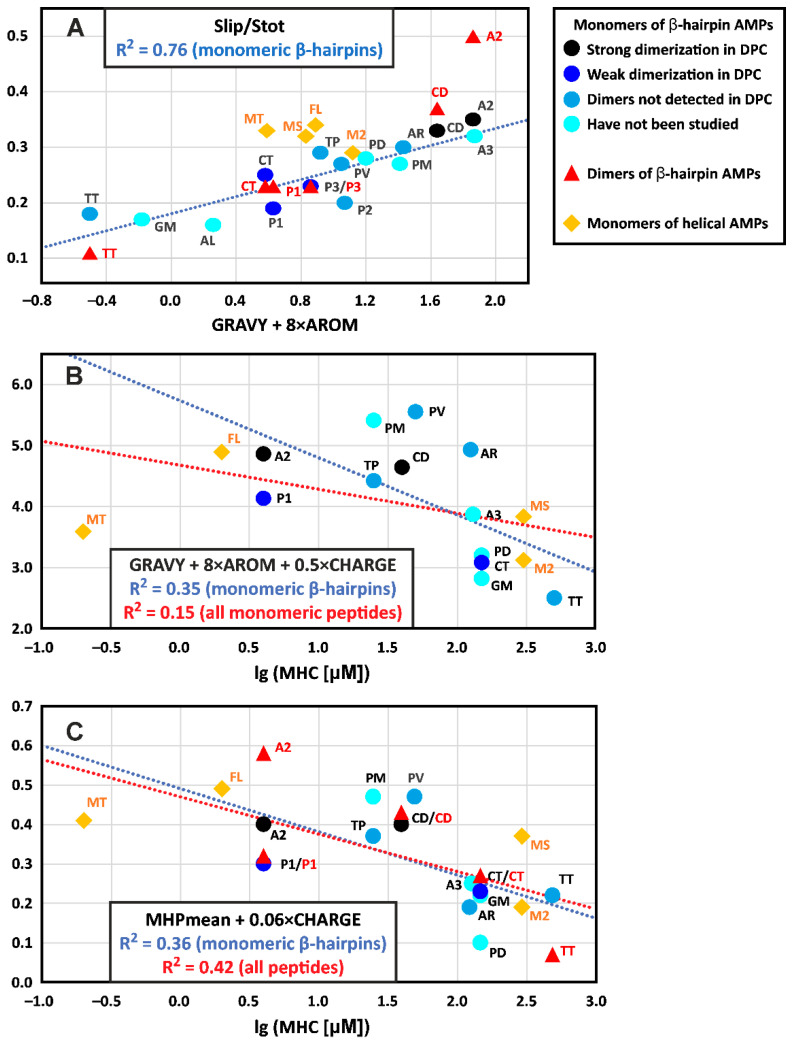
Correlations of the physicochemical properties of the β-hairpin and helical AMPs with their tendency to form dimers and hemolytic activity. (**A**) Relationship between the relative surface area of the hydrophobic regions (*S_lip_*/*S_tot_*) and the combination of the AMP hydrophobicity on the Kyte and Doolittle scale [28] (*GRAVY*) with the aromatic content (*AROM*). (**B**) Relationship between the combination of *GRAVY* and *AROM* parameters with the total charge of AMP (*CHARGE*) and its minimal hemolytic concentration (*MHC*). (**C**) Relationship between the combination of the average *MHP* value on the peptide surface (*MHP_mean_*) with the total charge and the *MHC* value. Abbreviations: Protegrin-3 (P3), Protegrin-2 (P2), Protegrin-1 (P1), Arenicin-2 (A2), ChDode (CD), Capitellacin (CT), Thanatin (TT), Tachyplesin I (TP), Arenicin-1 [V8R] (AR), PcDode (PD), Alvinellacin (AL), Polyphemusin I (PM), PV5 (PV), Arenicin-3 (A3), Gomesin (GM), Fowlicidin-1 (FL), Melittin (MT), Magainin-2 (M2), and MSI-594 (MS). To calculate the parameters of monomeric peptides, structures in DPC micelles were used. If they were not available, then the structures of peptides in water were used.

## Data Availability

All data generated or analyzed during this study are included in this article.

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
