# Peer review of "Dimerization of the β-Hairpin Membrane-Active Cationic Antimicrobial Peptide Capitellacin from Marine Polychaeta: An NMR Structural and Thermodynamic Study"

_biomolecules, 2024, doi:10.3390/biom14030332_

Round 1

Reviewer 1 Report

Comments and Suggestions for Authors

This manuscript entitled " Dimerization of the β-hairpin membrane-active cationic anti- microbial peptide capitellacin from the marine polychaeta. NMR structural and thermodynamics study” reports atomic-resolution structures of a 20-resiude cationic AMP capitellacin in DPC micelle. Authors also carried out PRE studies and thermodynamic analysis (NMR based) in their work. Structural Investigations of host defence peptides (HDP) or antimicrobial peptides (AMPs) could provide valuable insights developing peptide-based antibiotic analogs. At present, prevalence of MDR bacteria is challenging due to lack of new antibiotics. AMPs are pivotal leads in this regard. The current study is technically sound to using high-resolution NMR methods solving structures of capitellacin in a membrane mimic solution. In previous work [ref 20] authors have established recombinant production of the peptide and examined structure in water along with antibacterial activity. Capitellacin demonstrated potent activity against Gram-negative bacteria and lower toxicity to eukaryotic cells [ref 20]. Although the mode of action of the peptide remains unclear. Authors should address the following comments to improve the manuscript.

1.     What is the rational of selecting DPC as a membrane mimic conditions? DPC is neutral detergent and can better mimic eukaryotic membrane. Can the structures be correlated with low cytotoxicity/hemolytic activity of the AMP. Authors may look into structures of thanatin, MSI-594 AMPs in DPC micelle and potential correlations with lower cytotoxicity.

2.     What is the biological relationship of the monomeric/dimeric structures of Capitellacin with the antibacterial activity?

3.     Fig 2B HSQC shows multiple cross-peaks at the Gly box region (15N: 116-118 ppm) which is absent in Fig 2A. Authors may provide some explanation.

4.     Although amide proton temperature coefficient can provide information about H-bond, a complementary experiment H/D exchange may help in ascertaining the conclusion.

5.     Line 257: what is meant by pronounced amphipathicity?

6.     Oligomerization of AMPs are important for membrane permeabilization and disruption. Atomic-resolution structures of oligomeric AMPs in membrane like environment are limited. In order to better understand and broader impact in the field, authors should compare or mention atomic resolution structures of AMPs like thanatin, MSI-78, fowlicidin-1 (Biochim Biophys Acta. 2011 1808(1):369-81. doi: 10.1016/j.bbamem.2010.10.001).

Author Response

Reviewer #1

  1.    What is the rational of selecting DPC as a membrane mimic conditions? DPC is neutral detergent and can better mimic eukaryotic membrane. Can the structures be correlated with low cytotoxicity/hemolytic activity of the AMP. Authors may look into structures of thanatin, MSI-594 AMPs in DPC micelle and potential correlations with lower cytotoxicity.

Thank you for this question. Indeed, spherical micelles used for solution NMR studies are very dynamic in nature and highly mobile. They reproduce very poorly the charge distribution observed in real lipid bilayers. Using an anionic detergent to study a cationic AMP can result in the peptide aggregation, as shown in our recent study of ChDode in LMPG micelles [16]. Therefore, the zwitterionic detergents and especially dodecylphosphocholine (DPC) are traditionally used to study AMPs by NMR (see lot of references in the text of manuscript and references in Tables S4B-D in the supplementary materials). To ensure consistency with previous studies, we also used DPC for the current capitellacin study. To describe our motivation, we rewrote the beginning of 3.1 section in the manuscript (lines 236-255).

The second part of the question also requires complex answer. We analyzed the known data about structures and physicochemical properties of the β-hairpin peptides and available data about their hemolytic activity and, indeed, found such correlation. We found that dimers stability and hemolytic activity of β-hairpin AMPs are positively correlated with surface hydrophobicity. In addition, the positive correlation was observed between hemolytic activity and AMP charge. To describe these new findings, we added new figures (Figure 7 and Figure S4) and three new sections (4.1-4.3) in the Discussion of our manuscript (lines 458-655). These new results are also mentioned in the revised abstract and conclusions.

The description of Thanatin dimer and MSI-594 structure were added to the text of the manuscript and Supplementary Tables S4. Thanatin was also added to Figures 1 and 4. New references were added.

  1.    What is the biological relationship of the monomeric/dimeric structures of Capitellacin with the antibacterial activity?

This is a very interesting question. But now we do not have enough experimental data to firmly answer it. However, we have added a discussion of this issue to the manuscript (section 4.4, lines 732-750). Briefly. Our data (Fig. 7A) suggest that almost all β-hairpin AMPs can dimerize in the anionic environment of the bacterial membrane, but with varying efficiencies. We hypothesize that dimeric peptides have greater membrane and antimicrobial activity than monomeric peptides because dimeric peptides (excluding capitellacin) can form TM pores in the membrane, while monomeric peptides can only act through a carpet mechanism. Thus, a particular peptide can act on the membrane through a variety of possible mechanisms, depending on the state of dimer-monomer equilibrium under specific conditions. In this case, identifying the single mechanism responsible for the antimicrobial activity may be challenging. Likewise, measuring activity separately for the monomeric and dimeric forms of AMPs is also difficult.

  1.    Fig 2B HSQC shows multiple cross-peaks at the Gly box region (15N: 116-118 ppm) which is absent in Fig 2A. Authors may provide some explanation.

Thank you for this question. We did not pay sufficient attention to the description of this figure. We think that observed heterogeneity is linked with the cis–trans isomerization of the Ser1–Pro2 peptide bond in the dimer. This dipeptide belongs to the N-terminal β-strand responsible for the peptide dimerization, and changes in this site can cause rearrangement of the dimer structure. For the symmetric capitellacin dimer, three structurally different states are possible (cis/cis, cis/trans (equal to trans/cis), and trans/trans). The presence of the fourth component of the signal can be explained if we assume that the dimer symmetry is broken (cis/trans is not equal to trans/cis). We have changed the Fig 2AB and highlighted the additional signals for Val8, Gly20, and Hε1 Trp19 by circles. See the revised Figure 2 and its description in the manuscript (lines 267-282).

  1.    Although amide proton temperature coefficient can provide information about H-bond, a complementary experiment H/D exchange may help in ascertaining the conclusion.

Thank you for your remark. The H/D exchange and proton temperature coefficient are the two complimentary approaches to estimate the formation of hydrogen bonds in peptides. The experiment on H/D exchange requires the lyophilization of the sample, which may violate the monomer-dimer equilibrium in the sample (especially in the case then the part of the sample lost during lyophilization). Therefore, we prefer to obtain as many data as possible from one sample. The second reason is dynamics. Our peptide accomplishes several types of exchange motions in the mk-second to second timescale. These are the cis–trans isomerization of the Ser1–Pro2 peptide bond, monomer-dimer equilibrium by itself, and peptide binding from water phase to the micelle surface. In this case the H/D exchange can be very fast and cannot provide information about H-bonds.

  1.    Line 257: what is meant by pronounced amphipathicity?

We have removed “pronounced” from the text.

  1.    Oligomerization of AMPs are important for membrane permeabilization and disruption. Atomic-resolution structures of oligomeric AMPs in membrane like environment are limited. In order to better understand and broader impact in the field, authors should compare or mention atomic resolution structures of AMPs like thanatin, MSI-78, fowlicidin-1 (Biochim Biophys Acta. 2011 1808(1):369-81. doi: 10.1016/j.bbamem.2010.10.001).

Thank you for noting these works. We added a brief review of the MSI-78 and VK-22 (analogue of fowlicidin-1) dimers in the first paragraph of discussion section (lines 460-465). In the revised manuscript thanatin is discussed in the several places, in introduction and discussion sections, and it was added to Figures 1 and 4. Monomeric fowlicidin-1 and several available structures of thanatin were added to our analysis of AMPs (Supplementary Tables S4A-D).

Reviewer 2 Report

Comments and Suggestions for Authors

Antimicrobial resistance is increasing, and new antibiotics are needed. The manuscript by Mironov et al. describes the structural and functional characterization of the natural antimicrobial peptide capitellacin. Capitellacin forms a beta-hairpin structure, is membrane active, but its mode of action is not well understood. Likely the functional state of capitellacin is dimeric, as for other membrane active peptides. Using extensive NMR analysis, the authors determined the structure of the monomeric and dimeric capitellacin in DPC micelles, which is different from other dimeric antimicrobial peptides, and provide the thermodynamics of capitellacin dimerization. The authors identified the Trp19 residue as possibly functional relevant and provide a model how capitellacin might interact with membrane. The experiments are carefully performed, and the manuscript is well written.

I have only a minor comment. Only 10-15% of bacteria do contain phosphatidylcholine (DPC is a mimic for that). Instead, they contain phosphatidylethanolamine or cardiolipin with positively charged head groups, maybe it makes sense to also study capitellacin in this context.

Author Response

Reviewer #2

I have only a minor comment. Only 10-15% of bacteria do contain phosphatidylcholine (DPC is a mimic for that). Instead, they contain phosphatidylethanolamine or cardiolipin with positively charged head groups, maybe it makes sense to also study capitellacin in this context.

Thank you for your comment. To describe our motivation, we rewrote the beginning of 3.1 section in the manuscript (lines 236-255). We noted, that spherical micelles used for solution NMR studies are very dynamic in nature and highly mobile. They reproduce very poorly the charge distribution observed in real lipid bilayers. Using an anionic detergent to study a cationic AMP can result in the peptide aggregation, as shown in our recent study of ChDode in LMPG micelles [16]. Therefore, the zwitterionic detergents and especially dodecylphosphocholine (DPC) are traditionally used to study AMPs by NMR (see lot of references in the text of manuscript and references in Tables S4B-D in the supplementary materials). To ensure consistency with previous studies, we also used DPC for the current capitellacin study.

At the same time, lipid/detergent aggregates (bicelles) can also be used in solution NMR studies. Bicelles are better suited for modeling membrane properties. The use of bicelles with the addition of anionic lipids (PG or cardiolipin) is a good direction for our future studies of the interaction of β-hairpin AMPs with membranes.

Round 2

Reviewer 1 Report

Comments and Suggestions for Authors

All comments are adequately addressed.